# Feasibility of Overground Gait Training Using a Joint-Torque-Assisting Wearable Exoskeletal Robot in Children with Static Brain Injury

**DOI:** 10.3390/s22103870

**Published:** 2022-05-19

**Authors:** Juntaek Hong, Jongweon Lee, Taeyoung Choi, Wooin Choi, Taeyong Kim, Kyuwan Kwak, Seongjun Kim, Kyeongyeol Kim, Daehyun Kim

**Affiliations:** 1Department and Research Institute of Rehabilitation Medicine, Yonsei University College of Medicine, Seoul 03722, Korea; piyo1214@yuhs.ac (T.C.); kty745@yuhs.ac (T.K.); kskj9009@yuhs.ac (S.K.); hohoho7490@yuhs.ac (D.K.); 2Department of Physical Medicine and Rehabilitation, National Health Insurance Service Ilsan Hospital, Goyang 10444, Korea; bickpjl@yuhs.ac (J.L.); logoskyle@yuhs.ac (W.C.); 3Department of Rehabilitation Medicine, Gangnam Severance Hospital, Rehabilitation Institute of Neuromuscular Disease, Yonsei University College of Medicine, Seoul 03722, Korea; kyuwankw@yuhs.ac (K.K.); yed9411@yuhs.ac (K.K.)

**Keywords:** robot-assisted gait training, overground, joint-torque-assist, wearable robot, gait disturbance, children, static brain injury

## Abstract

Pediatric gait disorders are often chronic and accompanied by various complications, which challenge rehabilitation efforts. Here, we retrospectively analyzed the feasibility of overground robot-assisted gait training (RAGT) using a joint-torque-assisting wearable exoskeletal robot. In this study, 17 children with spastic cerebral palsy, cerebellar ataxia, and chronic traumatic brain injury received RAGT sessions. The Gross Motor Function Measure (GMFM), 6-min walk test (6 MWT), and 10-m walk test (10 MWT) were performed before and after intervention. The oxygen rate difference between resting and training was performed to evaluate the intensity of training in randomly selected sessions, while the Quebec User Evaluation of Satisfaction with assistive Technology 2.0 assessment was performed to evaluate its acceptability. A total of four of five items in the GMFM, gait speed on the 10 MWT, and total distance on the 6 MWT showed statistically significant improvement (*p* < 0.05). The oxygen rate was significantly higher during the training versus resting state. Altogether, six out of eight domains showed satisfaction scores more than four out of five points. In conclusion, overground training using a joint-torque-assisting wearable exoskeletal robot showed improvement in gross motor and gait functions after the intervention, induced intensive gait training, and achieved high satisfaction scores in children with static brain injury.

## 1. Introduction

Gait disability is one of the most common symptoms of neurological diseases [1]. Improving gait function in patients with gait disorders is the most important goal of rehabilitation. Gait disorders in children have various causes such as paralysis, abnormal muscle tone, and lack of muscle coordination [2]. These components are generally correlated in a complex manner. Moreover, pediatric patients have growth components unlike adult patients; therefore, pediatric gait disorders could be being complicated by combining various musculoskeletal problems, which challenge rehabilitation efforts.

Various rehabilitation treatment methods, including conventional physical therapy using physical effort, hydrotherapy, and electrical stimulation therapy, have been developed to improve gait ability in these patients. Among these treatments, robot-assisted gait training (RAGT) is a new rehabilitation treatment technique, the effectiveness of which has been reported by many studies [3,4,5]. The most well-known gait training robot, called a tethered robot, consists of a harness fixed to the ceiling and an exoskeleton fixed on a treadmill that provides a gait experience according to a pre-set gait trajectory. The outstanding advantage of this type of RAGT in previous studies was that it could provide repetitive and intensive experiences for the task of walking, which was possible only when two or more therapists intervened in conventional physical treatment [6,7].

However, only a few studies have examined the effects of RAGT in pediatric patients; thus, they remain unclear [8]. A previous study reported that a tethered-type exoskeletal robot could not facilitate muscle activity significantly since this type of robot only focused on achieving normal gait trajectories passively [9].

Alternatively, as technology develops, overground RAGT options that use wearable robots have been introduced that are not tethered and enable the participant to walk in various environments such as even levels as well as ramps or stairs. Overground RAGT using a wearable exoskeletal robot could improve patient’s active balance control, weight transfer, and muscle activation, and the reports have stated that overground gait training increases gait speed and endurance more than treadmill gait training [10,11]. However, studies of its feasibility and effectiveness on gait function are still lacking. There are only few studies [12,13] that analyzed the functional improvement after RAGT for overground walking in children, in contrast to RAGT with tethered robots [5]. This study aimed to determine the feasibility of overground RAGT in pediatric patients with static brain injury.

## 2. Materials and Methods

### 2.1. Design

This retrospective study was conducted at Severance Rehabilitation Hospital, Yonsei University College of Medicine. Ethical approval was granted by the institutional review board of our institute (1-2021-0090).

### 2.2. Participants

A total of 21 pediatric patients received overground RAGT as inpatients at our clinic between 1 January and 31 December 2021.

The inclusion criteria were as follows: (1) patients aged 6–18 years; (2) diagnosis of brain diseases for more than 6 months; (3) having received overground RAGT more than 10 times; (4) patients who could walk independently under supervision or could walk short distances with walking assistance corresponding to gross motor function classification system (GFMCS) level 2, 3, and 4; (5) sufficient cognitive function to follow the instructions of the training and evaluations (at least one step obey).

The exclusion criteria were as follows: (1) patients with intellectual disability to the extent that it was impossible to perform instructions during the training and evaluations; (2) having received orthopedic surgery or botulinum toxin injection within 3 months; (3) presence of lower extremity spasticity equal to or higher than 3, according to the modified ashworth scale.

As a result, we included 17 children in this study (9 males and 8 females).

### 2.3. Wearable Joint-Torque-Assisting Exoskeletal Robot

A wearable joint-torque-assisting exoskeletal robot, the Angel Legs M20 (ANGEL ROBOTICS Co., Ltd., Seoul, Korea), was used in this study. This robot can provide assistive torque according to the gait phase automatically detected by information combined from the ground contact sensor, encoders (incremental and absolute) in the actuators, and an inertial measurement unit sensor in a backpack (Figure 1). The joint actuators generated flexion torque at hip and knee joints during the swing phase and extension torque at hip and knee joints during the stance phase according to preformed polynomial assistive joint torque profile based on the gait phases. And the maximum points of flexion and extension torques of each joint could be set individually [14]. By utilizing a low-inertia motor and impedance reduction control algorithm, the robot can minimize discomfort and prevent unexpected resistance to a patient’s spontaneous movements. To optimize the wearer’s body size, the Angel Legs M20 consists of the following two types: the child-sized M20-A (dimensions: 390–510 mm [W] × 370–435 mm [D] × 1170–1430 mm [H]; weight, 14.2 kg) and the adult-sized M20-B (dimensions: 438–561 mm [W] × 425–480 mm [D] × 1284–1565 mm [H]; weight, 19.5 kg).

### 2.4. Robot-Assisted Gait Training Protocol for Pediatric Patients

The RAGT was conducted for 10–23 sessions (30 min per session, 4–5 sessions per week) for each participant. The first two to three sessions concentrated on fully adapting the participant to the state of wearing the robot, after which the interventional goals were reset based on the participant’s gait function and adaptation. If a participant could not walk independently while wearing a robot, the RAGT sessions were performed using a gait assist device such as a walker, a movable harness installed on the ceiling to prevent falls, and the assistance of a physical therapist. Meanwhile, if a participant could walk independently while wearing a robot, intensive gait training was provided to improve gait endurance and accommodate various environments such as ramps and stairs. All training sessions were conducted by a physical therapist with over 10 years of experience providing pediatric treatment. During the training period, the participants were required to maintain the same amount and level of conventional physical therapy (30 min once a day, 5 times a week) for as before starting intervention.

### 2.5. Clinical Evaluation

Primary outcome measures were: (1) gross motor function using the Gross Motor Function Measure (GMFM); (2) gait ability using the 6-min walk test (6 MWT) and 10-m walk test (10 MWT) without a robot before and after the RAGT sessions were performed.

Secondary outcome measures were: (1) oxygen consumption during randomly selected training sessions to analyze the intensity of training; (2) information about side effects was collected through interviews with the study staff after each session. If an adverse event occurred, the symptoms were recorded on the case report form; (3) satisfaction assessment of the gait-assistive device, the Quebec User Evaluation of Satisfaction Assistive Technology 2.0 (QUEST 2.0)-Korean version [15]. The schematic flow of the protocol is shown in Figure 2.

### 2.6. Gross Motor Function Measure

The GMFM is a standardized assessment tool used to evaluate the gross motor function of patients with cerebral palsy. Moreover, the GMFM has acceptable validity for other pediatric patients with neurological abnormalities [16]. To assess gross motor function, the total GMFM score and all five dimensions (lying and rolling; sitting; crawling and kneeling; standing; walking, running, and jumping) were measured. The sum of the items in each dimension is indicated as a percentage [17].

### 2.7. Six-Minute Walk Test

All the patients underwent this test to evaluate gait endurance. This assessment tool has been proven safe, easy to perform, and highly acceptable in children [18]. During the 6 MWT, the distance (in meters), walked every minute for 6 min, was measured. The use of any gait assistive device was permitted, and each patient used the same device in all assessment sessions. In total, 11 of 17 patients in this study used gait-assist device during the assessment. Only four patients used ankle-foot orthosis, one patient used only walker, and six patients used both ankle-foot orthosis and walker. If a participant was unable to complete the 6-min walk, only the distance walked within the time was measured.

### 2.8. Ten-Meter Walk Test

The 10 MWT is a reliable and valid assessment tool for evaluating locomotor capacity [19]. In this study, the gait speed when walking for 10 m at a comfortable speed and maximum speed was calculated. When measuring comfortable speed, the evaluator asked the participants to walk according to the usual walking speed. In contrast, when measuring maximal speed, the evaluator asked the participants to walk as fast as they safely can. Gait speed was measured in m/s.

### 2.9. Oxygen Consumption

The oxygen rate was recorded using a wireless metabolic analyzer (K4b2; Cosmed, Rome, Italy) to analyze the intensity of exercise. The wireless metabolic analyzer has the advantage of being easy to use during overground training due to its portability and being lightweight. For each participant, one session was randomly selected, and the oxygen rate was measured. After the child rested in the seated position for 15 min, the oxygen rate was measured. The maximal oxygen rate during the first 5 min of the 30-min training session was measured.

### 2.10. QUEST 2.0-Korean Version

The QUEST 2.0, which evaluates patient satisfaction with various assistive technologies, has proven reliability, validity, and applicability [20]. This tool consists of eight domains. These are effectiveness, comfort, ease of use, durability, safety, adjustment, weight, and dimension. All eight domains were evaluated on a 5-point scale, and the last question involved choosing the three most outstanding domains. After the entire training session, the questionnaire was completed by each participant himself/herself, but for the parts that were difficult to understand, the caregivers helped. The data were collected in September 2021.

### 2.11. Statistics

The statistical analysis was performed using the SPSS software (version 26.0; SPSS Inc., Chicago, IL, USA). When the normality test was performed, all values except the oxygen rate did not satisfy the normal distribution. Therefore, the Wilcoxon signed-rank test was used to compare the changes in pre-and post-training GMFM, 10 MWT, and 6 MWT. A paired *t*-test was used to compare the changes in the oxygen rate between the resting and training states. Statistical significance was set at *p* < 0.05.

## 3. Results

### 3.1. Patient Characteristics

Only 17 of 21 patients satisfied the inclusion criteria and were included in this study (Figure 3). The general characteristics of the participating children are presented in Table 1. The etiologies of the brain injury in this study included 12 patients with spastic cerebral palsy, 4 patients with ataxic quadriplegia and 1 patient with chronic traumatic brain injury incurred more than 6 months prior.

Some of the patients could not complete the assessment according to the protocol; a schematic diagram of these assessments, including the RAGT protocol, was shown in Figure 3. Two patients could not complete the 6 MWT and 10 MWT since they could not walk without the assistance of a robot, and one patient refused to perform the 6 MWT (Figure 3).

### 3.2. Changes between Pre- and Post-Training

In the GMFM, all domains—lying and rolling; sitting; crawling and kneeling; standing, walking, and running; jumping—improved significantly (*p* < 0.05) except for the lying and rolling domain (Table 2). In the 6 MWT, distances measured every minute and total walking distance for 6 min were longer than those in the pre-training evaluation (*p* < 0.05) (Table 3). Moreover, both comfortable walking speed and maximum walking speed after training were significantly faster than those of the pre-training 10 MWT (*p* < 0.05) (Table 4).

### 3.3. Changes in Physiologic Burden between Resting and Training States

All patients showed a significantly higher oxygen rate during the training when compared to resting state (*p* < 0.05). In particular, the oxygen rate was 200% higher than the baseline rate in 14 of 17 patients (maximally 600%) (Figure 4).

### 3.4. Satisfaction Questionnaire

In total, six children among the participants completed the QUEST 2.0, with a median total score of 34.0 of 40. The median values for each domain are listed in Table 5. The domain that received the highest score was effective, and the other domains that received a high score of 4 or more were durability, safety, dimensions, ease of use, and comfort. However, the weight and adjustment domains scored less than 4 points.

In the selection of the best rating domains, the most common selection was effectiveness, followed by comfort, ease of use, and safety. Durability, adjustment, weight, and dimensions were selected by less than half of the participants. The details are shown in Figure 5.

### 3.5. Adverse Events

Three adverse events occurred among the participants. These were discomfort, fatigue, and musculoskeletal pain. However, all symptoms disappeared within 30 min of symptom onset, and there were no serious adverse events such as skin injury, fall, sprain, or fracture.

## 4. Discussion

This is the first study to analyze the feasibility of an overground, joint-torque-assisting wearable robot for children with long-term gait disabilities. The 6 MWT, 10 MWT, and GMFM measurements showed significant improvement in the second assessment when compared to the first assessment. The degree of exercise intensity estimated by the physical burden was also significantly higher than that at rest. In addition, high satisfaction scores were obtained.

In some previous studies with patients with cerebral palsy, which did not use an exoskeletal device, overground gait training increased gait speed and endurance more than treadmill-based gait training [10,11]. Moreover, overground gait training using a wearable robot was demonstrated as more effective for dynamic balance than tethered-type robots [21,22]. They emphasized that facilitation of the trunk muscles for appropriate alignment of the body axis during weight-shifting movement could improve both static and dynamic balance control, followed by gait function [23,24]. Considering that gross motor function reflects dynamic balance [25,26], the simultaneous improvement of gross motor function and gait ability through RAGT was shown in this study. In addition, the joint-torque-assisting control method induced a more synergetic effect than overground training. As shown in a previous study, the joint-torque-assisting control method created a more stable force output than the passive method [27]. This control method could provide assistance more accurately according to patient intent [28]. Providing repetitive movements accurately and reflecting the patient’s intentions could promote motor development after brain injuries [29]. Besides, enjoying the training program could improve gait function [30]. Since there was no control group in this study, it was difficult to compare the effects of RAGT and conventional physical therapy. In addition, several variables during training, such as torque amplitude, therapist assistance intensity, and whether weight-bearing tools and walking aids are used, could influence the interventional effect. In a future study, we intend to perform a more standardized RAGT protocol by controlling for these variables and comparing RAGT with conventional treatment.

Intensive gait training is an important factor of gait rehabilitation [31]. Thus, the point worth emphasizing in this study is to make the attempt to measure the exercise intensity during the gait training. The majority of previous studies analyzed the intensity by measuring the changes in heart rate during RAGT [9,32]. However, by measuring the oxygen rate using a metabolic gas analyzer, a more accurate analysis was achieved in this study. The significant increase in the oxygen rate, which value was between comfortable speed and fast speed walking in able-bodied children of similar age [33], implied that RAGT induced intensive gait training, even in young children. On the other hand, six patients among the participants had past RAGT history using a different type of robot, Lokomat (V5.0; Hocoma AG; Volketswil, Switzerland), within 1 year and oxygen consumption values using the same protocol during the session. It showed significantly lower oxygen rate differences compared to overground RAGT using the Angel Legs M20-A. Moreover, a previous study analyzing the oxygen demand during RAGT showed that a joint-torque-assisting wearable robot induced a higher oxygen consumption rate than a joint-trajectory-controlled wearable robot (17.21 ± 3.74 mL/kg/min versus 11.2 ± 1.7 mL/kg/min, respectively) [34]. However, a well-designed further study would be needed for quantitative comparison among the different types of robots.

This result could be explained by the slacks of the model [35], the fundamental property of the human motor system that continuously attempts to reduce the effort when a repetitive movement error is small, which can be a factor that interferes with rehabilitation efforts. Tethered RAGT systems with a constant predetermined trajectory are vulnerable to slacking. In contrast, the joint-torque-assisting method could be less likely to “slack” by facilitating one’s own power to walk in addition to the external assistance power provided by the actuator. In addition, children who could not walk independently experienced significant physical burdens. They may have never experienced the extent of the burden, which exerts a positive effect on the musculoskeletal and cardiopulmonary areas. Thus, it may be necessary to analyze the effect of an appropriate degree of physical burden in further studies.

It is noteworthy that participant satisfaction with the safety of RAGT were clearly shown in this study. No major adverse events occurred, similar to the results of other studies on the safety of overground wearable robots. The satisfaction analysis using the QUEST 2.0 showed a high score in this study. However, the weight and adjustment domains showed relatively low scores than other domains. Overcoming these challenging points through the development of lightening hardware and weight distribution techniques [36] were considered critical points to increase user acceptability and interventional effectiveness in the future. In particular, Shingarey et al., argued that the joint-torque-assisting method could be better at improving the human-robot interaction than other control methods which reflects the potential of this control method [37], which shows the development potential of this control method for RAGT. To ensure an accurate comparison of pediatric patients in the future, satisfaction analyses will have to use a more child-oriented satisfaction evaluation tool with more samples, such as the QUEST 2.1 Children’s Version [38].

In conclusion, gross motor and gait functions significantly improved after overground RAGT using a joint-torque-assisting wearable exoskeletal robot in children with static brain injury. In addition, this overground RAGT induced intensive gait training with high safety and satisfaction.

## Figures and Tables

**Figure 1 sensors-22-03870-f001:**
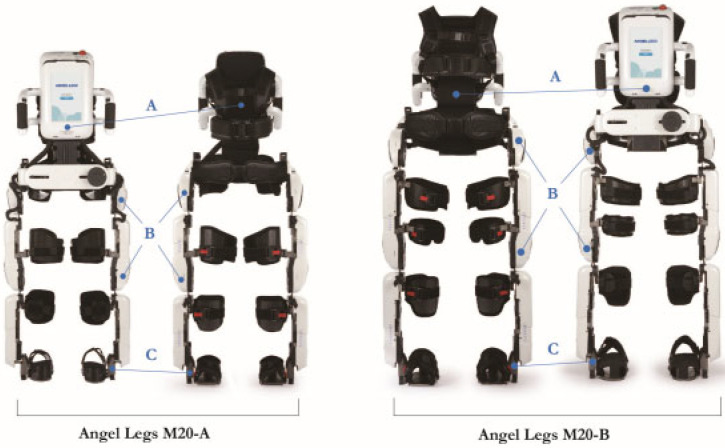
Joint-torque-assisting Wearable exoskeletal robot. The Angel Legs-M20 consists of a child-sized model (the M20-A) and an adult-sized model (the M20-B): A: a controller backpack that includes a battery and inertial measurement unit sensor in the backpack circuit for analyzing trunk tilt angle; B: hip and knee joint actuator to guide walking posture by generating energy as well as an incremental encoder for joint angular velocity measurement installed in the actuator motor circuit and an absolute encoder to measure the absolute joint angle in the output shaft of the actuator; C: ground contact sensor at the lateral side of the foot support.

**Figure 2 sensors-22-03870-f002:**
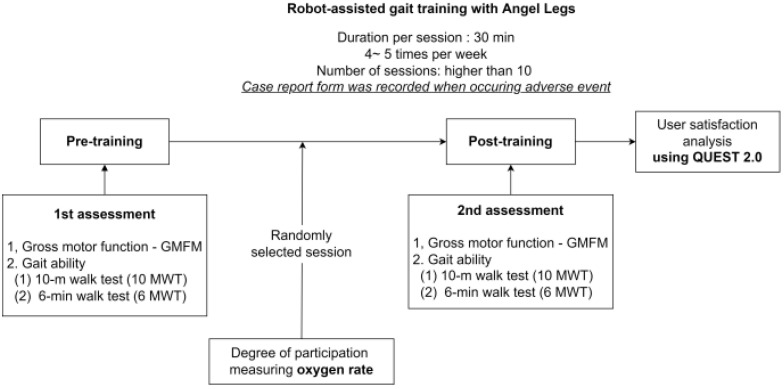
Robot-assisted gait training protocol using the Angel Legs for pediatric patients. GMFM, gross motor function measurement; QUEST 2.0, Quebec User Evaluation of Satisfaction with assistive Technology 2.0.

**Figure 3 sensors-22-03870-f003:**
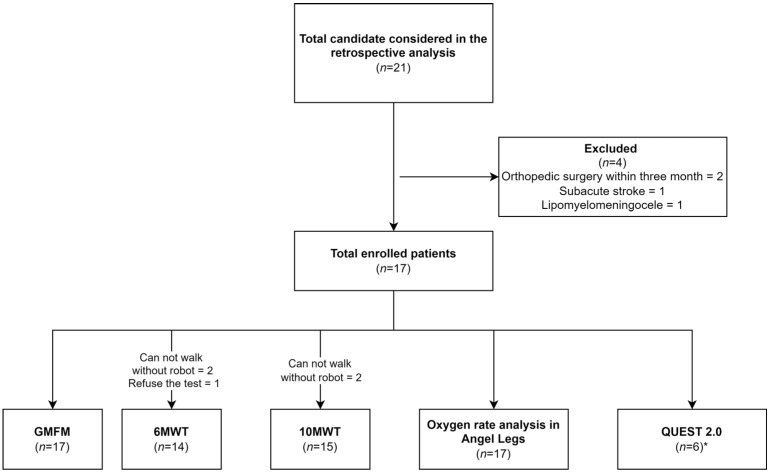
Schematic diagram of the evaluation, including the robot-assisted gait training protocol in this study. * Low sample size owing to the starting period of this evaluation tool; 6 MWT, 6-min walk test; 10 MWT, 10-m walk test; GMFM, Gross Motor Function Measures; QUEST 2.0, Quebec User Evaluation of Satisfaction with assistive Technology 2.0.

**Figure 4 sensors-22-03870-f004:**
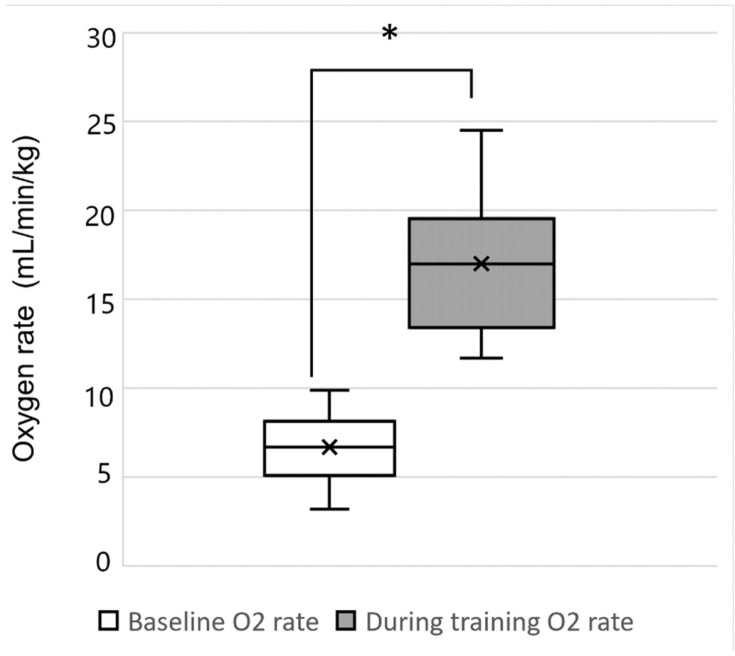
Oxygen rate of the resting when compared to resting state. Data were analyzed using paired *t*-test. Values are shown as mean ± SD. * *p* < 0.05.

**Figure 5 sensors-22-03870-f005:**
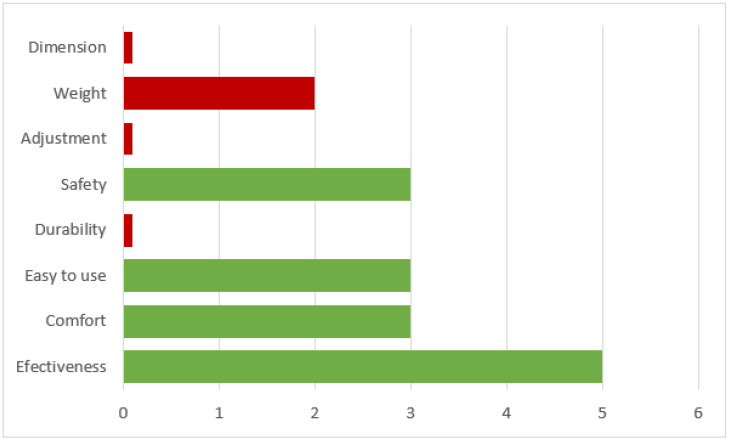
Results of the Quebec User Evaluation of Satisfaction with assistive Technology 2.0 test. The score of each topic refers to the number of participants who considered that characteristic relevant. (Green indicates more than half of the participants, while red indicates less than half of the participants.).

**Table 1 sensors-22-03870-t001:** General characteristics of the participants.

	Value
Total patients	17
Age, years	9.40 (8, 17)
Sex, M:F	9:8
Height, cm	136.00 (116.6, 172.0)
Weight, kg	36.00 (22.7, 59.0)
Training session	20.0 (10, 23)
Diagnosis	
Cerebral palsy, spastic type	12
Ataxic quadriparesis	4
Traumatic brain injury	1

Values are shown as *n* or median (minimum, maximum).

**Table 2 sensors-22-03870-t002:** GMFM scores at pre-and post-training time points.

	Pre-Training	Post-Training
Lying and rolling	100.0 (100.0, 100.0)	100.0 (100.0, 100.0)
Sitting	100.0 (65.0, 100.0)	100.0 (70.0, 100.0) *
Crawling and kneeling	86.9 (9.5, 100.0)	94.1 (11.9, 100.0) *
Standing	57.7 (2.6, 94.9)	69.2 (12.8, 94.9) *
Walking, running, and jumping	49.3 (0.0, 97.2)	50.0 (6.9, 97.2) *
Total GMFM	78.86 (35.41, 98.41)	82.38 (49.26, 98.41) *

Data were analyzed using the Wilcoxon signed-rank test. Values are presented as median (minimum, maximum). * *p* < 0.05. GMFM, Gross Motor Function Measure.

**Table 3 sensors-22-03870-t003:** Six-minute walk test distances at pre-and post-training time points.

Distance at Time	Pre-Training	Post-Training
1 min	35.45 (1.58, 63.52)	43.21 (7.50, 120.30) *
2 min	77.28 (3.16, 121.10)	95.54 (13.50, 186.90) *
3 min	117.58 (4.72, 193.80)	145.09 (17.20, 278.30) *
4 min	155.15 (6.40, 257.92)	190.87 (21.00, 341.34) *
5 min	188.22 (7.82, 337.14)	239.43 (26.40, 425.82) *
6 min	223.10 (9.48, 415.50)	284.28 (30.80, 500.12) *

Data were analyzed using the Wilcoxon signed-rank test. Values are presented as medians (minimum, maximum). * *p* < 0.05.

**Table 4 sensors-22-03870-t004:** Ten-meter walk test distances at pre-and post-training time points.

	Pre-Training	Post-Training
10 MWT (comfortable speed, m/s)	0.49 (0.03, 1.03)	0.69 (0.13, 2.40) *
10 MWT (maximal speed, m/s)	0.68 (0.03, 1.51)	0.91 (0.14, 2.55 ) *

Data were analyzed using the Wilcoxon signed-rank test. Values are shown as median (minimum, maximum). * *p* < 0.05.

**Table 5 sensors-22-03870-t005:** Mean Quebec User Evaluation of Satisfaction with assistive Technology 2.0 scale scores.

	Median (Minimum, Maximum)
Dimensions	4.0 (4.0, 5.0)
Weight	3.0 (2.0, 4.0)
Adjustment	3.0 (3.0, 5.0)
Safety	4.5 (3.0, 5.0)
Durability	4.5 (4.0, 5.0)
Ease of use	4.0 (3.0, 5.0)
Comfort	4.0 (2.0, 5.0)
Effectiveness	5.0 (3.0, 5.0)
Total	34.0 (25.0, 37.0)

## Data Availability

The data presented in this study are available upon request from the corresponding authors. The data are not publicly available for reasons related to participant privacy.

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
