# Peer review of "Feasibility of Overground Gait Training Using a Joint-Torque-Assisting Wearable Exoskeletal Robot in Children with Static Brain Injury"

_sensors, 2022, doi:10.3390/s22103870_

Round 1
Reviewer 1 Report
I have the following editorial comments. Thanks for addressing my previous comments
- In the abstract in lines 59-60, “There are only few study analyzed the functional improvement after RAGT with overground type in children [11,12].” I recommend replacing “There are only few studies that analyzed the functional improvement after RAGT for overground walking in children [11,12]”.
- In the abstract in sentences 61-62, “This feasibility study aimed to determine the feasibility of 61 overground RAGT in pediatric patients with static brain injury.” Delete the first “feasibility” to precent repetition.
- In sentences 169-170 “The oxygen rate was recorded using a wireless metabolic analyzer (K4b2; Cosmed, Rome, Italy) to analyze the intensity of exercise which reflects muscle activity during the RAG.” I recommend deleting “which reflects muscle activity during the RAG”. This could be confusing since muscle activity could be measured more explicitly using EMG.
Author Response
Dear reviewer
We sincerely appreciate the your comments that proved valuable in improving the quality of our paper. We have addressed all the comments and revised our manuscript accordingly. Our responses to all the comments and the major revisions are listed below.
Point 1: In the abstract in lines 59-60, “There are only few study analyzed the functional improvement after RAGT with overground type in children [11,12].” I recommend replacing “There are only few studies that analyzed the functional improvement after RAGT for overground walking in children [11,12]”.
Response 1: Thank you for pointing out the details. I modified the sentence as you pointed out
(Lines in 59-61)
Before : There are only few study analyzed the functional improvement after RAGT with overground type in children
After : There are only few studies that analyzed the functional improvement after RAGT for overground walking in children
Point 2: In the abstract in sentences 61-62, “This feasibility study aimed to determine the feasibility of 61 overground RAGT in pediatric patients with static brain injury.” Delete the first “feasibility” to precent repetition.
Response 2: Thank you for pointing out what I hadn't thought of. I deleted the first “feasibility” from the sentence.
(Lines in 61-62)
Before : This feasibility study aimed to determine the feasibility of 61 overground RAGT in pediatric patients with static brain injury
After : This study aimed to determine the feasibility of overground RAGT in pediatric patients with static brain injury
Point 3: In sentences 169-170 “The oxygen rate was recorded using a wireless metabolic analyzer (K4b2; Cosmed, Rome, Italy) to analyze the intensity of exercise which reflects muscle activity during the RAG.” I recommend deleting “which reflects muscle activity during the RAG”. This could be confusing since muscle activity could be measured more explicitly using EMG.
Response 3 : Thank you for this comment. For clarifying the meaning, I modified the sentence you pointed out
(Lines in 169-170)
Before : The oxygen rate was recorded using a wireless metabolic analyzer (K4b2; Cosmed, Rome, Italy) to analyze the intensity of exercise which reflects muscle activity during the RAGT
After : The oxygen rate was recorded using a wireless metabolic analyzer (K4b2; Cosmed, Rome, Italy) to analyze the intensity of exercise.

Reviewer 2 Report
Dear Authors, in this resubmission, I did not find a significant improvement in the Manuscript, when compared to the old version.
Specifically, most of the previous weaknesses still remain, specifically:
- the different kinds of disease;
- the different rehabilitation doses among participants (10–23 sessions).
Shifting from efficacy to a feasibility study lowered the impact of the above-mentioned limits, but what about feasibility? Besides the adverse events, the only feasibility measure (i.e., the Quebec User Evaluation of Satisfaction) was collected in only 6 over 17 participants. Moreover, I disagree with the comparison with other studies in the discussion, where very different cohorts of patients were enrolled.
Author Response
Dear reviewer
We sincerely appreciate the your comments that proved valuable in improving the quality of our paper. We have addressed all the comments and revised our manuscript accordingly. Our responses to all the comments and the major revisions are listed below.
Point 1:
Dear Authors, in this resubmission, I did not find a significant improvement in the Manuscript, when compared to the old version. Specifically, most of the previous weaknesses still remain, specifically:
- the different kinds of disease;
- the different rehabilitation doses among participants (10–23 sessions).
Response 1: Thank you for this comment. We are very sorry that you mentioned there is no significant improvement, especially the different kinds of disease and intervention doses. In order to reduce heterogeneity of the disease group in this study, we excluded the spinal cord injury group and analyzed for the static brain injury group more than 6 months after diagnosis. Neverthless there had been heterogeneity such as disease group and the number of interventions because this study was designed for a retrospective study.
Based on this feasibility study, our study group just started a multi-center prospective study to prove the rehabilitative effect of this joint-torque-assisting wearable exoskeletal robot in children with spastic CP. It would be appreciated if you could take this into consideration.
Point 2: Shifting from efficacy to a feasibility study lowered the impact of the above-mentioned limits, but what about feasibility? Besides the adverse events, the only feasibility measure (i.e., the Quebec User Evaluation of Satisfaction) was collected in only 6 over 17 participants. Moreover, I disagree with the comparison with other studies in the discussion, where very different cohorts of patients were enrolled.
Response 2 : Thank you for this comment. It is considered the limitation of this study that only a small number of satisfaction analysis samples were included. Also, I agreed that it's not appropriate to compare other studies because the sample size was too small for comparison. Considering your point, the comparison part has been modified as shown below.
Before (line in 321-327 before the modification)
The satisfaction analysis using the QUEST 2.0 showed a high score in this study. Compared to other overground type devices, the total score was higher than others such as the EKSO GT® (31.3 ± 5.70) [34] and the ReWalk Rehabilitation 2.0 (29.36 ± 2.48) [35] which are joint-trajectory-controlled wearable robots, as well as the Marsi Active Knee® (22.4 ±3.2) [36] another joint-torque-assisting control type. However, compared to the tethered robot, the total score was relatively low, particularly that of the weight and adjustment domains
After (line in 322-323 before the modification)
The satisfaction analysis using the QUEST 2.0 showed a high score in this study. However, the weight and adjustment domains showed relatively low scores than other domains.

Round 2
Reviewer 2 Report
I would like to thank the Authors for having, at least partially, considered my comments.
- However, I further suggest to better report the main limits of the research (the retrospective nature of the study, the heterogeneity of the sample, the heterogeneity of the intervention in terms of dose, and the lack of a control group) at the end of the discussion section;
Moreover, I suggest to consider the suggestions reported below.
- in the abstract, the retrospective nature of the study should be declared;
- the term “enrolled” should be avoided across the manuscript, since it is a retrospective study;
- line 36: “Moreover, pediatric patients have growth components unlike adult patients; therefore, pediatric gait disorders are often chronic and accompanied by various complications, which challenge rehabilitation efforts”. What is the link between the growth and the chronicity?
- Line 41: Among these treatments, robot-assisted gait training (RAGT) is a new rehabilitation treatment technique, the effectiveness of which has been reported by many studies [3,4]. Information on which patient populations should be added. A very important paper on the topic is missing (10.1016/j.apmr.2019.08.479). I suggest to improve the references.
- Line 46: “The outstanding advantage of this type of RAGT in previous studies was its repeatability and task-oriented characteristics, which was possible only when two or more therapists intervened in conventional physical treatment”. The sentence is unclear. I suggest to rephrase it.
- Line 50: “A previous study reported that a tethered-type exoskeletal robot could reduce muscle activity as these kinds of robots focus only on achieving normal gait trajectories and cannot reflect the patient's gait intention”. The sentence is unclear. I suggest to rephrase it.
- Line 59: “There are only few studies that analyzed the functional improvement after RAGT for overground walking in children”. A very important paper on the topic is missing (10.1016/j.apmr.2019.08.479). I suggest to improve the references, especially those related to the use of RAGT in children.
- Line 92: "according to preformed assistive torque profile which could be individually set". Please better specify what preformed assistive torque profile are.
- Line 119: “During the training period, the participants were required to maintain the same amount and level of conventional physical therapy as before starting intervention”. Please better describe the type and amount of conventional physical therapy.
- Line 157: "The participant was allowed to finish the walk". What does it mean?
- Line 162: "Determination of speed was conducted in such a way that the evaluator verbally asked the participants during the evaluation". The sentence is unclear. I suggest to rephrase it.
- Line 174: "The maximal oxygen rate during the first 5 min of the 30-min training session was measured". Why were the first 5 minutes selected? Please explain.
- Figure 4: please revise it, numbers are in Korean. Remove table, since the same data are reported.
- Line 262: "However, all symptoms disappeared within 30 min after the end of intervention". Did you mean the end of the session?
- Line 271: "which was higher than that of the tethered robots". What does this sentence refer to?
- Line 285: "This high quality of human-machine adaptability could produce more precise repetitive movements and may promote motor development after brain injuries”. The sentence is unclear. I suggest to rephrase it.
- Line 295. The authors interpreted positively the results about the oxygen consumption. First, it seems trivial to me that it is higher during training, when compared to rest. Moreover, in authors’ opinion, how to distinguish a beneficial and an excessive intensive training, based on this measure? Please comment on that.
- Lines 295-317: the authors did not compare tethered vs overground RAGT, then these paragraphs are actually not supported by data. Please amend them.
- Line 327: "which shows the development potential of this control method for RAGT". The sentence is unclear. I suggest to rephrase it.
Finally, I suggest to revise the language of the manuscript, especially the novel parts (the highlighted parts). As an example (not exhaustive), please amend the following sentences/words:
- Please use 6-minute walk test, instead of 6-min walking test (and the acronym 6MWT, instead of 6 MWT);
- Please use 10-meter walk test, instead of 10-m walking test (and the acronym 10MWT, instead of 10 MWT);
- Line 56: improve a patient’s active balance control
- Rephrase the third inclusion criteria, according to the previous one (e.g., “presence of lower extremity spasticity equal to or higher than 3, according to the modified ashworth scale”);
- Line 83: As a result, we enrolled 17 children (9 males and 8 females patients).
- For primary outcome measures Primary outcome measures were:
- Secondary outcome measures were performed as the following, Secondary outcome measures were:
- Semicolon instead of point right before “(3) satisfaction assessment” (line 132);
- Figure 2: Number of sessions: higher than 10 Total session: more that 10 times:
- GMFM scores: please round to one decimal place;
- Table 3: Six-MWT 6-min walking test
- during the training versus resting state, when compared to resting state
- Figure 4: Oxygen rate of the resting versus training statuses. Rephrase.
- Line 243: Six children from among the enrolled participants
- Line 269: in the pre-versus post- 269 training periods. Rephrase.
- Line 197: Most previous studies The majority of previous studies
- Line 303: higher oxygen consumption rate
Author Response
Dear reviewer
I sincerely appreciate your comments that proved valuable in improving the quality of our paper. I have addressed all the comments and revised our manuscript accordingly. The consent of all authors was obtained for the revised contents. Since the content is too large, I have attached a word file for detailed information. Thanks again for your hard work.

Round 3
Reviewer 2 Report
We would like to thank the Authors for having considered my suggestion.
I only further suggest a final English editing from someone with full professional proficiency in English.
This manuscript is a resubmission of an earlier submission. The following is a list of the peer review reports and author responses from that submission.
Round 1
Reviewer 1 Report
Review on “Effects and feasibility of overground gait training using a wearable torque-assisted wearable exoskeletal robot in children with various gait disabilities”
- Rewrite the title to account to eliminate the repeated word “wearable”
- It is recommended to replace the word “treatment” with “intervention”
- In the abstract lines 24-25, the following “All items except weight had higher satisfaction scores than the other robots” is ambiguous as the “other robots” have not been properly introduced.
- What is the main objective of enrolling participants with different neurological conditions?
- In lines 66-67, the following “However, no comprehensive research has compared which control method has a superior rehabilitative effect” is confusing, because the reviewer would expect a study design to generate data to provide comparisons between controllers (e.g., position-based controller versus torque-based controller).
- In lines 77-78, it is confusing how 21 participants received overground walking training and then 18 were enrolled. Did 3 participants do not complete the study?
- What motor or sensory assessments are conducted to determine the walking ability of the participants prior to enrollment?
- In lines 95-97, be careful on the conventions with flexion and extension for hip and knee joints, respectively.
- Provide a reference or explain the design of the control algorithm.
- In lines 155-156, what is meant by gait assistive device? Canes, walkers, ankle-foot orthoses? Please be specific.
- In linear 162-163, it is confusing to describe that participants walked at a comfortable speed and maximum speed? What is the difference? What instructions were provided to participants to segregate between comfortable and fast walking?
- In lines 188-189, what is the rationale to use a paired t-test compared to the non-parametric Wilcoxon signed ranked test (that is used for the other outcome measures)?
- In the abstract and results section, care must be exercised to prevent skewing the conclusions as it is confusing to make comparisons (oxygen rate differences) between the overground walking and treadmill training when only 6 participants (out of the total participants) had previous experience with the Lokomat. Isn’t this a conflictive factor for participant enrollment since this influence the inclusion/exclusion criteria?
Author Response
Dear reviewer
We sincerely appreciate the your comments that proved valuable in improving the quality of our paper. We have addressed all the comments and revised our manuscript accordingly. Our responses to all the comments and the major revisions are listed below.
Point 1: Rewrite the title to account to eliminate the repeated word “wearable”
Response 1: Thank you for this comment. We apologize for the duplicate words being included in the title. We will fix that as follows.
Before
Effects and feasibility of overground gait training using a wearable torque-assisted wearable exoskeletal robot in children with various gait disabilities
After
Effects and feasibility of overground gait training using a wearable torque-assisted exoskeletal robot in children with various gait disabilities
Point 2: It is recommended to replace the word “treatment” with “intervention”
Response 2: Thank you for this comment. All of the word ‘treatment’ associated with RAGT mentioned in this study have been replaced with the word ‘intervention’. And we highlited the modified part.
Point 3: In the abstract lines 24-25, the following “All items except weight had higher satisfaction scores than the other robots” is ambiguous as the “other robots” have not been properly introduced.
Response 3: Thank you for pointing out what I hadn't thought of. As you know, direct comparison with other robots such as tethered-type device(Lokomat) or position-based unthetered-type device (EKSO GT®, ReWalk Rehabilitation 2.0) could not be possible because there was no control group in this study. So we compared indirectly satisfaction analysis scores with other types of wearable robots referring to other studies. So it may be inappropriate to put comparisons with other types of robots in the abstract part.
As your advice, I would like to modify the sentence as follows.
Before
All items except weight had higher satisfaction scores than the other robots
After
All items except weight had higher satisfaction scores
Point 4: What is the main objective of enrolling participants with different neurological conditions?
Response 4: Thank you for your valuable comment. The purpose of this study is not focused on the effect of RAGT in a specific patient group. In line with the purpose of the retrospective and feasibility study, we would like to analyze the comprehensive aspect of RAGT in children diagnosed with various neuromuscular diseases.
Meanwhile, as you pointed out, the group of participants were very heterogenous. It seems possible to clarify the group as 'static brain injury' by excluding participant diagnosed with lipomyelomeningocele. I would like to ask your valuable opinion.
Point 5: In lines 66-67, the following “However, no comprehensive research has compared which control method has a superior rehabilitative effect” is confusing, because the reviewer would expect a study design to generate data to provide comparisons between controllers (e.g., position-based controller versus torque-based controller).
Response 5: Thank you for your comment. The expression was not appropriate. So, I changed the expression as follows (Line in 66-68)
Before
However, no comprehensive research has compared which control method has a superior rehabilitative effect
After
However, there are few study analyzed the rehabilitative effect of RAGT with torque-based control system. Moreover, to my knowledge, there have been only case studies on the effectiveness of torque-assisted wearable robot in children
Point 6: In lines 77-78, it is confusing how 21 participants received overground walking training and then 18 were enrolled. Did 3 participants do not complete the study?
Response 6: I’m very sorry for the insufficient explanation. Actually, among 21 pediatric patients received overground torque-assisted RAGT, 3 patients were excluded because they didn’t satisfy the inclusion criteria. Detailed information about the reasons why the inclusion criteria were not satisfied was written in Figure 3. In order to understand this more easily, I will add this information in the part ‘3.1. Patient characteristics’ (Line 210-211)
Point 7: What motor or sensory assessments are conducted to determine the walking ability of the participants prior to enrollment?
Response 7: I’m very sorry for the insufficient explanation. we used the gross motor function classification system (GFMCS) for classfying the walking ability and we enrolled the patients who could walk independently under supervision or could walk short distances with walking assistance GFMCS level 2, 3 and 4). we will add this items to the inclusion criteria in lines with 84-86
Before
The inclusion criteria were as follows: (1) patients aged 6–18 years; (2) diagnosis of brain or spinal cord diseases for more than 6 months; (3) having received overground torque-assisted RAGT more than 10 times; (4) sufficient cognitive function to follow the instructions of the training and evaluations.
After
The inclusion criteria were as follows: (1) patients aged 6–18 years; (2) diagnosis of brain or spinal cord diseases for more than 6 months; (3) having received overground torque-assisted RAGT more than 10 times; (4) patients who could walk independently under supervision or could walk short distances with walking assistance corresponding to gross motor function classification system (GFMCS) level 2, 3 and 4; (5) sufficient cognitive function to follow the instructions of the training and evaluations (at least one step obey).
Points 8 : In lines 95-97, be careful on the conventions with flexion and extension for hip and knee joints, respectively.
Point 9: Provide a reference or explain the design of the control algorithm.
Response 8 & 9: Thank you for this comment. In the stance phase, an extension moment is provided to both the knee and hip joint, and in the swing phase, a flexion moment is provided to both the knee and hip joint. We modified the sentence in lines with 101-104 for clarifying the meaning. And then we add the reference about the method of providing torque during the gait cycle.
Before
The joint actuators generated flexion torque during the swing phase and extension torque during the stance phase at the hip and knee joints.
After
The joint actuators generated flexion torque at hip and knee joints during the swing phase and extension torque at hip and knee joints during the stance phase according to preformed assistive torque profile which could be individually set [12].
Adding reference
Choi, H. Assistance of a Person with Muscular Weakness Using a Joint-Torque-Assisting Exoskeletal Robot. Applied Sciences 2021, 11, doi:10.3390/app11073114.
Point 10: In lines 155-156, what is meant by gait assistive device? Canes, walkers, ankle-foot orthoses? Please be specific.
Response 10: Thank you for your comment. 12 of 18 patients used gait-assist device during the assessment. 4 patients only used ankle-foot orthosis, 1 patient used only walker and 7 patients used both ankle-foot orthosis and walker. I will add this information in lines 163-165.
Point 11: In linear 162-163, it is confusing to describe that participants walked at a comfortable speed and maximum speed? What is the difference? What instructions were provided to participants to segregate between comfortable and fast walking?
Response 11: Thank you for your comment. It seems that we did not clearly provide the difference between maximal speed and comfortable speed. As you know, according to the principle of measuring 10MWT* (We will put the reference below the paragraph), it is recommended to measure at both comfortable and maximal speed. Especially in this study, all participants enrolled in this study had sufficient cognitive function to follow the instructions of evaluations, determination of speed was conducted in such a way that the evaluator verbally asked the participants during the evaluation. When measuring comfortable speed, the evaluator asked the participants to walk according to the usual walking speed. In contrast, when measuring maximal speed, the evaluator asked the participants to walk as fast as they safely can. I will add an explanation for that part in lines 172-176.
*Reference
- Pirpiris, M., Wilkinson, A., et al. "Walking speed in children and young adults with neuromuscular disease: comparison between two assessment methods." Journal of Pediatric Orthopaedics 2003 23(3): 302
- Watson, M. J. "Refining the ten-metre walking test for use with neurologically impaired people." Physiotherapy 2002 88(7): 386-397
- Steffen, T. and Seney, M. "Test-retest reliability and minimal detectable change on balance and ambulation tests, the 36-item short-form health survey, and the unified Parkinson disease rating scale in people with parkinsonism." Physical Therapy 2008 88(6): 733-746
Point 12: In lines 188-189, what is the rationale to use a paired t-test compared to the non-parametric Wilcoxon signed ranked test (that is used for the other outcome measures)?
Response 12 : When the normality test was performed, the oxygen rate value satisfies the normal distribution, unlike other measured values. So the parametric test, paired-t test, was performed only for the oxygen rate value. I will add an explanation for this in lines 203-204.
Point 13: In the abstract and results section, care must be exercised to prevent skewing the conclusions as it is confusing to make comparisons (oxygen rate differences) between the overground walking and treadmill training when only 6 participants (out of the total participants) had previous experience with the Lokomat. Isn’t this a conflictive factor for participant enrollment since this influence the inclusion/exclusion criteria?
Response 13 : Thank you for your comment. Considering only 6 of participants had experience with treadmill-based RAGT, it could be a conflictive factor as you pointed out. However, we would be very appreciated if you consider this study is a feasibility study of the torque-assisted wearable robot.
In the process of collecting data to analyze the feasibility of the werable robot used in this study, we found some data from the previous RAGT using LOKOMAT, so we wanted to share these interesting results . As you pointed out, these results could not guarantee superiority. In the future, well-organized study will be needed.
Considering your points, we modified the expressions likely to skew the conclusion in the abstract and conclusion parts (lines in 26 and 368)

Reviewer 2 Report
Aim of this paper is to assess effects and feasibility of a commercial exoskeleton for gait training of children. The authors test the robotic system on a cohort of 21 pediatric subjects with different gait disabilities. In the introduction, the authors describe two types of gait training devices, the tethered or the overground robots. Moreover, they describe the difference between position and torque based controls. While tethered-robots have limitation that induce to prefer the overground choose, the authors point out that there is not a comprehensive research comparing the two control methods and claiming the best. Therefore the objective of is to test effects and feasibility of an overground torque-based control exoskeleton for gait training.
The authors use standard clinical tests, such as GMFM, 6 MWT and 10 MWT, as measures of the effects of training. They express feasibility in terms of level of satisfaction using the device, with the QUEST 2.0 inventory. Further measure of training effect, provided by the authors, is the patient participation; this expressed in terms of metabolic expenditure. Statistics is simply based on looking for differences pre vs post training.
In general, results show an improvement in in the clinical evaluation and an increase in the oxygen rate. The level of satisfaction is high. With these results, the authors conclude that overground training using a torque-assisted exoskeleton robot are feasible and have good effects on patients.
The study can be of interest for the person in the field. However, studying only the torque-based control exoskeleton without a comparison with a position-based control system is a limit of the paper. Moreover, there are some important aspects do not result appropriate for what the authors assert in the conclusion. In particular, they risk carrying the wrong messages about what robotics in rehabilitation is and how it works. Please find my major comments below:
Q1: If the authors point out there is not a comprehensive research that compares which of the two control methods has best rehabilitative effects, why the authors evaluate only the effects of a torque-based control robot?
Q2: With respect to the literature, could the authors explain what is innovative in their results, comparing the tethered (LOKOMAT) and the overground (Angel Legs M20) robots? Why did the authors show this result, but did not describe it in the methods section?
Q3: In my opinion, the authors looking at the oxygen rate and interpreting it as participation, are falling in several errors.
- Participation means the patient’s involvement in performing a motor task that is goal-oriented. This type of task has been proven to have a major rehabilitative effect with respect the movement without a purpose. But there is not, at the extent of my knowledge, a direct link between the increase of oxygen rate and participation.
- The increase of oxygen rate would represent an increase of muscles activity. Therefore, if the authors consider robots as rehabilitative devices that only ensure an elevated number of repetition of the same movement, the increase of the metabolic expenditure is a clear sign of success of the machine. However, honestly, this would be in opposite to the concept of participation and goal-oriented rehabilitative protocols. Therefore, it is reductive towards rehabilitation robotics.
- Moreover, please take attention, more effort could not mean more positive effects in training.
Q4: 10 MWT and 6 MWT are quantitative measures; why did the authors use a non-parametric test? Did the authors studied the distributions of the measures?
Minor comments:
Line259: is the durability score reported in table 5 correct?
Line 344: is the reference correct?
Author Response
Dear reviewer
We sincerely appreciate the your comments that proved valuable in improving the quality of our paper. We have addressed all the comments and revised our manuscript accordingly. Our responses to all the comments and the major revisions are listed below.
Point 1: If the authors point out there is not a comprehensive research that compares which of the two control methods has best rehabilitative effects, why the authors evaluate only the effects of a torque-based control robot?
Response 1: Thank you for pointing out what I hadn't thought of. As stated in this paper, there are two control methods of wearable robots. Among them, there are relatively many studies on the effects of the position-based controlled wearable robot such as the EKSO GT® and the ReWalk Rehabilitation 2.0 mentioned in this study. In contrast, few gait-assist robots with torque-assisted controlled method have been introduced for rehabilitation so far. To our knowledge, there is only one, HAL® Exoskeleton. In addition, there have been only case studies on the effectiveness of torque-assist control RAGT focused on pediatric patients. So we would like to analyze the feasibility of torque-assisted controlled werable robot.
Reflecting what you pointed out, we will add the information about this in lines with 66-68
Point 2: With respect to the literature, could the authors explain what is innovative in their results, comparing the tethered (LOKOMAT) and the overground (Angel Legs M20) robots? Why did the authors show this result, but did not describe it in the methods section?
Response 2: I appreciate your valuable comment. In the process of collecting data to analyze the feasibility of the werable robot used in this study, we found some data from the previous RAGT using LOKOMAT, so we would like to share these interesting results.
As you pointed out, we will add this information in the methods section(lines in 187-189)
Additional information
If there was past RAGT history using a different type of robot within 1 year and oxygen consumption values using the same protocol during the session. The comparison of the oxygen rate difference values were also performed between robots.
Point 3 In my opinion, the authors looking at the oxygen rate and interpreting it as participation, are falling in several errors. Participation means the patient’s involvement in performing a motor task that is goal-oriented. This type of task has been proven to have a major rehabilitative effect with respect the movement without a purpose. But there is not, at the extent of my knowledge, a direct link between the increase of oxygen rate and participation.
The increase of oxygen rate would represent an increase of muscles activity. Therefore, if the authors consider robots as rehabilitative devices that only ensure an elevated number of repetition of the same movement, the increase of the metabolic expenditure is a clear sign of success of the machine.
However, honestly, this would be in opposite to the concept of participation and goal-oriented rehabilitative protocols. Therefore, it is reductive towards rehabilitation robotics.
Moreover, please take attention, more effort could not mean more positive effects in training.
Response 3: Thank you for pointing out what I hadn't thought of. There might be a logical leap by confusing the degree of energy consumption with the participation.
However, it is clear that increase of the metabolic expenditure, that can be expressed as ‘active engagement’, during the RAGT in this study. It could be proven by the slack model mentioned in this paper; the fundamental property of the human motor system that continuously attempts to reduce the effort when a repetitive movement error is small. And the this may partially reflect the participation. However, it cannot be denied that the expression ‘the participation’ was exaggerated in this paper. To accurately measure the degree of participation, further study will be needed.
Considering your points, we will change the expression from participation to active engagement, or suggest possibility to reflect participation. (Lines in 19, 26, 138, 296, 321-334, 368)
Point 4: 10 MWT and 6 MWT are quantitative measures; why did the authors use a non-parametric test? Did the authors studied the distributions of the measures?
Response 4: I’m very sorry for the insufficient explanation. When the normality test was performed, the oxygen rate value satisfies the normal distribution, unlike other measured values. So the parametric test, paired-t test, was performed only for the oxygen rate value. I will add an explanation for this in lines 203-204..
Point 5: Line259: is the durability score reported in table 5 correct?
Response 5: I checked the part again and confirmed that there were no errors.
Point 6:Line 344: is the reference correct?
Response 6: Thank you for pointing out the parts that haven't been corrected. There was an error in the reference, so we corrected the error (line in 365).

Reviewer 3 Report
Thank you for the opportunity to review this manuscript. The topic was: Effects and feasibility of overground gait training using a wearable torque-assisted wearable exoskeletal robot in children with various gait disabilities.
This preliminary clinical study aimed to determine the feasibility and effect of overground torque-assisted RAGT on gait function in pediatric patients with various neurological diseases.
The manuscript is interesting, but I have a few comments to the authors:
The study has a small number of participants, in addition to suffering from various neurological diseases, this group has certain limitations in terms of the pattern of gait characteristic for a given disease.
Please describe the clinical condition of the participants in terms of their functional status, level of spasticity and cognitive functions.
Was the cognitive state assessed according to one accepted criterion?
Were the participants' ability to speak and communicate assessed?
Was the functional status assessed, especially of the lower limbs, was the level of spasticity assessed, and if so what scale was used?
Author Response
Dear reviewer
We sincerely appreciate the your comments that proved valuable in improving the quality of our paper. We have addressed all the comments and revised our manuscript accordingly. Our responses to all the comments and the major revisions are listed below.
Point 1: Please describe the clinical condition of the participants in terms of their functional status, level of spasticity and cognitive functions.
Was the cognitive state assessed according to one accepted criterion?
Were the participants' ability to speak and communicate assessed?
Was the functional status assessed, especially of the lower limbs, was the level of spasticity assessed, and if so what scale was used?
Response 1
I’m very sorry for the insufficient explanation. To evaluate their functional status, we used the gross motor function classification system (GFMCS) for classfying the walking ability and we enrolled the patients who can walk independently under supervision or can walk short distances with walking assistance (Gross motor function classification system (GFMCS) level 2,3 and 4). And we enrolled participants who could follow the instructions of the training and evaluations (at least 1 step obey). Moreover, participants with grade 3 or higher lower extremity spasticity were excluded by using the modified ashworth scale. we will add these informations to the inclusion criteria and exclusion criteria in lines with 82-93

Reviewer 4 Report
The authors analyzed data from 18 pediatric patients about the effects of an overground exoskeleton in their study.
The topic is intriguing; however, the presence of too many biases, such as:
- the heterogeneity of the sample (both clinically, and in terms of the provided intervention - 10 to 23 sessions);
- the comparison with a different treatment (using a thetered device) without any information about this treatment;
- the retrospective nature of the study;
- the lack of data in 3 patients
preclude drawing any meaningful conclusions from the study results.
Author Response
Dear reviewer
We sincerely appreciate the your comments that proved valuable in improving the quality of our paper. We have addressed all the comments and revised our manuscript accordingly. Our responses to all the comments and the major revisions are listed below.
The topic is intriguing; however, the presence of too many biases, such as:
Point 1: the heterogeneity of the sample (both clinically, and in terms of the provided intervention - 10 to 23 sessions);
Response 1 : Thank you for your valuable comment. In accordance with the purpose of the feasibility study, we attempted to compare the effects by registering as many patients as possible. So we agree that the data could be heterogenous. However, to our knowledge, there have been only case studies on the effectiveness of torque-assisted wearable robot in children. And we are conducting a prospective study to clarify the effect of RAGT using torque-assisted werable. We would appreciate it if you could consider these aspects.
Point 2: the comparison with a different treatment (using a thetered device) without any information about this treatment;
Response 2 : I appreciate your comment. It seems that the description of treatment using a tethered device was omitted. In the process of collecting data to analyze the feasibility of the werable robot used in this study, we found some data of oxygen rate from the previous RAGT using LOKOMAT, so we would like to share these interesting results.
As you pointed out, additional information about treatment using a tethered robot was added to the methods section. (lines in 187-189)
Additional information
If there was past RAGT history using a different type of robot within 1 year and oxygen consumption values using the same protocol during the session. The comparison of the oxygen rate difference values were also performed between robots.
Point 3: the retrospective nature of the study
Response 3: Thank you for your comment. As stated in this paper, there were only few gait-assist robots with torque-assisted controlled method. To our knowledge, there is only one for rehabilitative purpose, HAL® Exoskeleton. In addition, there have been only case studies on the effectiveness of torque-assist control RAGT focused on pediatric patients. So we would like to analyze the feasibility of torque-assisted controlled werable robot. We would be very appreciated if you consider this study is a feasibility study of this torque-assisted wearable robot.
Point 4: the lack of data in 3 patients
Response 4: I’m very sorry for the insufficient explanation. Actually, among 21 pediatric patients received torque-assisted werable RAGT, 3 patients were excluded because they didn’t satisfy the inclusion criteria. Detailed information about the reasons why the inclusion criteria were not satisfied was written in Figure 3. In order to understand this more easily, I will add this information in the part ‘3.1. Patient characteristics’ (Line 210-211)

Round 2
Reviewer 1 Report
Thanks for the responses to my previous comments. I have the following feedback
- The abstract still requires a lot of work.
- The objective of the study is not clear, and this is confusing the reviewers. Is the objective only to establish feasibility? Then say so in the abstract. If you are going to determine the effects, then the study population must be more narrowed down to make sense from the obtained results.
- What are the main outcome measures (select 2 and stick to them)? Clarify the selected primary and secondary measures. Then, in the main body of the paper you can talk about other measures, if needed. Broadening too much the measures induces confusion.
- You performed an overground robot-assisted gait training intervention. But somehow you have “The oxygen rate difference was significantly higher than 23 treadmill-based passive-type RAGT in several participants.” Are you then comparing overground versus treadmill training? Is that the objective of the study, then why mentioning the comparison? However, if your objective in this study is to compare the effects of overground walking in children versus treadmill walking then clarify that.
- (From my question in the previous round of review), I believe this issue has not been addressed. In lines 24-25, it is written “All items except weight had higher satisfaction scores.” I think it is even more ambiguous now. You use the word “higher”, but with respect to what?? If this sentence is not contributing to the main objective, then delete it. Or describe that a secondary analysis was done to compare those “items” between different devices.
- I appreciate the effort on clarifying the objective of enrolling participants with different neurological conditions. Based on your response you say “The purpose of this study is not focused on the effect of RAGT in a specific patient group. In line with the purpose of the retrospective and feasibility study, we would like to analyze the comprehensive aspect of RAGT in children diagnosed with various neuromuscular diseases.” So, the reviewer is again confused, and it relates to the main issue that the objective of the study is not clear. Why the use the word “Effects” in the title. If the study is a retrospective and feasibility study. Then, clarify that that is the case.
- I appreciate the effort on clarifying my question in the previous review on the torque control system. However, the response provided by the authors is at best confusing. So, are the authors saying that testing the torque-based controller is also a contribution? What is it implied by “Rehabilitative effect” and “Effectiveness” in line 66-68? There are many RAGT with torque-based control systems that have been examined for gait speed, endurance, etc. Also, in the opinion of the reviewer care must be taken because that paragraph in the introduction that discusses the control systems is very dangerous. The most confusing point is that inherently ALL exoskeleton controllers applied “torque assistance.” Now how those torque-based exoskeletons achieve that can vary (e.g., impedance, admittance, direct torque control, etc.). Is it relevant for the study to discuss such nuances? If the torque-based controller is not a major contribution, then why confuse the readers with the differences in the control systems. Now, if the goal of the study was to compare controller A vs. controller B during overground walking, then mentioning the control systems would be of paramount importance.
- Be careful with the edits applied as some of the new language is confusing, in first person (Which is different from what you had in the first submission)
Reviewer 2 Report
Oxygen consumption represents that muscles are more involved/engaged. Writing that bigger muscles work can be interpreted as sign of learning is in my opinion an inappropriate speculation.
Actually muscles activity can represents participation in a specific context that imply for instance training while executing a motor learning task in which the patient acquires skills. Therefore I suggest the authors to be careful in confusing muscles activity as indicator of participation. To better understand what I’m meaning I suggest the authors to read:
Krishnan C, Ranganathan R, Dhaher YY, Rymer WZ (2013) A Pilot Study on the Feasibility of Robot-Aided Leg Motor Training to Facilitate Active Participation. PLoS ONE 8(10): e77370. doi:10.1371/journal.pone.0077370
It is clear that this study is not a similar case.
Reviewer 3 Report
Thank you the authors for the improved version of the manuscript, I am satisfied
Reviewer 4 Report
Dear Authors,
I would like to thank you for the changes made to the text, which have made it more readable. However, my previous concerns were not adequately addressed, as reported below.
Reply to Response 1. I continue to believe that combining data from a sample that is too heterogeneous (both clinically and in terms of the intervention provided) will not allow significant inferences to be drawn.
Reply to Response 2. Lines 187-189 refers to the description of the 10 MWT. Moreover, "history using" should be better defined.
Reply to Response 3. "We would be very appreciated if you consider this study is a feasibility study of this torque-assisted wearable robot", but in the title the word effects is present.
Reply to Response 4. I was referring to patients who could not complete the assessment according to the protocol.